# Detecting Label Errors in Token Classification Data

## Abstract

Mislabeled examples are a common issue in real-world data, particularly for tasks like token classification where many labels must be chosen on a fine-grained basis. Here we consider the task of finding sentences that contain label errors in token classification datasets. We study 11 different straightforward methods that score tokens/sentences based on the predicted class probabilities output by a (any) token classification model (trained via any procedure). In precision-recall evaluations based on real-world label errors in entity recognition data from CoNLL-2003, we identify a simple and effective method that consistently detects those sentences containing label errors when applied with different token classification models.

## 1 Introduction

In most machine learning (ML) settings, datasets used for training and validation are typically assumed to be error-free. However, label errors are increasingly prevalent in real-world datasets (Northcutt et al., 2021a). One study found that datasets labeled by third-party annotators can contain between 7% to 80% label errors (Northcutt et al., 2021a). The presence of label errors introduces two problems: training on noisy labels can deteriorate the model accuracy, while testing on noisy labels can make the results less reliable.

The task of label error detection (LED) has shown promising results in higher model accuracy (Kuan & Mueller, 2022). To improve the dataset independent of the model, an approach is to first detect potential label errors, then manually review those errors and train the model using the cleaned dataset (Northcutt et al., 2021b). In this paper, we apply the same workflow on named entity recognition (NER) tasks and focus on different methods to effectively detect label errors. While existing LED benchmarks typically introduce artificial label errors, in which labels are randomly swapped with other labels (Northcutt et al., 2021b), such analyses do not resemble real-world data since annotators are more likely to make mistakes at a different distribution that is infeasible to model (Wei et al., 2022). Therefore, we benchmark our LED algorithm using CoNLL-2003, which has been used extensively in NER research.

It has recently come to light that many supervised learning datasets contain numerous incorrectly labeled examples (Northcutt et al., 2021a). To efficiently improve the quality of such data, Label Error Detection (LED) has emerged as a task of interest (Kuan & Mueller, 2022; Northcutt et al., 2021b; Müller & Markert, 2019), in which algorithms flag examples whose labels are likely wrong for reviewers to inspect/correct. This may be done by means of a score for each example which reflects its estimated *label quality*. A useful score ranks mislabeled examples higher than others, allowing reviewers to much more efficiently identify the label errors in a dataset.

This paper considers LED for token classification tasks (such as *entity recognition*) in which each token in a sentence has been given its own class label. While it is possible to score the labels of individual tokens, reviewing a candidate token flagged as potentially mislabeled requires looking at the entire sentence to understand the broader context. Here we propose `worst-token`, a method to score sentences based on the likelihood that they contain some mislabeled tokens, such that sentences can be effectively ranked for efficient label review[1]. We evaluate the LED performance of this approach and others on real-world data with naturally occuring label errors, unlike many past LED evaluations based on synthetically-introduced

---

[1]Code for our method is available here: `https://github.com/anonymized`

label errors (Brodley & Friedl, 1999; Müller & Markert, 2019; Gu et al., 2021), for which conclusions may differ from real-world errors (Kuan & Mueller, 2022; Jiang et al., 2020; Wei et al., 2022).

## 2 Related Work

Extensive research has been conducted on standard classification with noisy labels (Song et al., 2022; Wei et al., 2022; Chen et al., 2019; Müller & Markert, 2019; Natarajan et al., 2013; Brodley & Friedl, 1999). Our work builds on label quality scoring methods for classification data studied by Northcutt et al. (2021b); Kuan & Mueller (2022), which merely depend on predictions from a trained multiclass classification model. These methods are straightforward to implement and broadly applicable, being compatible with any classifier and training procedure, as is our `worst-token` method.

Only a few prior works have studied label error detection for token classification tasks specifically (Wang et al., 2019; Reiss et al., 2020; Klie et al., 2022). Also aiming to score the overall label quality of an entire sentence like us, Wang et al. (2019) propose `CrossWeigh` which: trains a large ensemble of token classification models with *entity-disjoint* cross-validation, and scores a sentence based on the number of ensemble-member class predictions that deviate from the given label, across all tokens in the sentence. Reiss et al. (2020) propose a similar approach to LED in token classification, which also relies on counting deviations between token labels and corresponding class predictions output by a large ensemble of diverse models trained via cross-validation. Unlike our `worst-token` method, the methods of Wang et al. (2019); Reiss et al. (2020) are more computationally complex due to ensembling many models, and do not account for the confidence of individual predictions (as they are based on hard class predictions rather than estimated class probabilities). Given the LED performance of `worst-token` improves with a more accurate token classifier, we expect `worst-token` to benefit from ensembling's reliable predictive accuracy improvement in the same way that ensembling benefits the methods of Wang et al. (2019); Reiss et al. (2020). We also tried entity-disjoint vs. standard data splitting, but did not observe benefits from the former variant (it often removes a significant number of entities).

Klie et al. (2022) study various approaches for LED in token classification, but only consider scores for individual tokens rather than entire sentences. Here we compare against some of the methods that performed best in their study. The studies of Klie et al. (2022); Northcutt et al. (2021b); Kuan & Mueller (2022) indicate that label errors can be more effectively detected by considering the confidence level of classifiers rather than only their hard class predictions. Instead of depending on predicted class probabilities for each token, our `worst-token` method appropriately accounts for classifier confidence.

## 3 Methods

Typical token classification data is composed of many sentences (i.e. training instances), each of which is split into individual tokens (words or sub-words) where each token is labeled as one of $K$ classes (i.e. entities in entity recognition). Given a sentence $x$, a trained token classification model $M(\cdot)$ outputs predicted probabilities $\mathbf{p} = M(x)$ where $p_{ij}$ is the probability that the $i$th token in sentence $x$ belongs to class $j$. Throughout, we assume these probabilities are *out-of-sample*, computed from a copy of the model that did not see $x$ during training (e.g. because $x$ is in test set, or cross-validation was utilized).

Using $p$, we first consider evaluating the individual per-token labels. Here we apply effective LED methods for standard classification settings (Northcutt et al., 2021b) by simply treating each token as a separate independent instance (ignoring which sentence it belongs to). Following Kuan & Mueller (2022), we compute a label quality score $q_i \in [0,1]$ for the $i$th token (assume it is labeled as class $k$) via one of the following options:

- self-confidence (`sc`): $q_i = p_{ik}$, i.e. predicted probability of the given label for this token.

- normalized margin (`nm`): $q_i = p_{ik} - p_{i\tilde{k}}$ with $\tilde{k} = \arg\max_j \{p_{ij}\}$

- confidence-weighted entropy (`cwe`): $q_i = \dfrac{p_{ik}}{H(p_i)}$  where $H(p_i) = -\dfrac{1}{\log K} \sum_{j=1}^{K} p_{ij} \log(p_{ij})$

Higher values of these label quality scores correspond to tokens whose label is more likely to be correct (Kuan & Mueller, 2022). We can alternatively evaluate the per-token labels via the Confident Learning algorithm of Northcutt et al. (2021b), which classifies each token as correctly labeled or not ($b_i = 1$ if this token is flagged as likely mislabeled, $= 0$ otherwise) based on adaptive thresholds set with respect to per-class classifier confidence levels.

For one sentence with $n$ word-level tokens, we thus have:

- **p**, a $n \times K$ matrix where $p_{ij}$ is model predicted probability that the $i$th token belongs to class $j$.

- **l** $= [l_1, \ldots, l_n]$, where $l_i \in \{0, \ldots, K-1\}$ is the given class label of the $i$th token.

- **q** $= [q_1, \ldots, q_n]$, where $q_i$ is a label quality score for the $i$th token (one of the above options).

- **b** $= [b_1, \ldots, b_n]$, where $b_i = 1$ if $i$th token is flagged as potentially mislabeled, otherwise $b_i = 0$.

Recall that to properly verify whether a token is really mislabeled, a reviewer must read the full sentence containing this token to understand the broader context. Thus the most efficient way to review labels in a dataset is to prioritize inspection of those *sentences most likely to contain a mislabeled token*. We consider 11 methods to estimate an overall quality score $s(x)$ for the sentence $x$, where higher values correspond to sentences whose labels are more likely all correct.

1. `predicted-difference`: The number of disagreements between the given and model-predicted class labels over the tokens in the sentence, also utilized in the methods of Wang et al. (2019); Reiss et al. (2020). Here we break sentence-score ties in favor of the highest-confidence disagreement. More formally:
$$s(x) = -|\mathcal{R}| - \max_{i \in \mathcal{R}} p_{i, \hat{l}_i}$$
where $\hat{l}_i = \arg\max_j \{p_{ij}\}$ and $\mathcal{R} = \{i : \hat{l}_i \neq l_i\}$. If $\mathcal{R} = \varnothing$, we let $\max_{i \in \mathcal{R}} p_{i, \hat{l}_i} = 0$.

2. `bad-token-counts`: $s(x) = -\sum_i b_i$, the number of Confident Learning flagged tokens. Similarly considered by Klie et al. (2022), this approach is a natural token-classification extension of the method of Northcutt et al. (2021b) for LED in standard classification tasks.

3. `bad-token-counts-avg`: Again scoring based on number of tokens flagged as potentially mislabeled, but now breaking ties primarily via the average label quality score of the flagged tokens and secondarily via the average label quality score of the other tokens. More formally:
$$s(x) = -\sum_i b_i + \frac{1}{|\mathcal{R}|} \sum_{i \in \mathcal{R}} q_i + \frac{\epsilon}{|\mathcal{S}|} \sum_{i \in \mathcal{S}} q_i$$
where $\mathcal{R} = \{i : b_i = 1\}$, $\mathcal{S} = \{i : b_i = 0\}$, and $\epsilon$ is some small constant.

4. `bad-token-counts-min`: Similar to `bad-token-counts-avg`, but break ties using minimum token quality rather than average token quality. More formally:
$$s(x) = -\sum_i b_i + \min_{i \in \mathcal{R}} q_i + \epsilon \cdot \min_{i \in \mathcal{S}} q_i$$

5. `good-fraction`: Fraction of tokens not flagged as potential issues, $s(x) = -\dfrac{1}{n} \sum_{i=1}^{n} b_i$.

6. `penalize-bad-tokens`: Penalize flagged tokens based on their corresponding label quality scores. More formally,

$$s(x) = 1 - \frac{1}{n} \sum_{i=1}^{n} b_i(1 - q_i)$$

7. `average-quality`: Average label quality of tokens in the sentence, $s(x) = \frac{1}{n} \sum_{i=1}^{n} q_i$.

8. `product`: $s(x) = \sum_i \log(q_i + c)$, where $c$ is a constant hyperparameter. This score places greater emphasis on tokens with low estimated label-quality, while still being influenced by all tokens' quality (like the previous `average-quality` method). With $q$ based on `sc` or `nm` token-scores, the `product` and `average-quality` methods are natural sentence extensions of the CU or PM methods considered in Klie et al. (2022) for token-level LED.

9. `expected-bad`: A rough approximation of the expected number of mislabeled tokens in sentence. More formally:

$$s(x) = \sum_{j=1}^{\min(n,J)} j \cdot q^{(j)}$$

where $q^{(i)}$ is the $i$th lowest token label quality score in this sentence, and $J$ is a hyperparameter. If using the `sc` label-quality score, $1 - q^{(i)}$ can be considered a loose proxy for the probability of having at least $i$ label errors in this sentence.

10. `expected-alt`: Similar to `expected-bad`, but only considering the likelihood of any label error rather than how many might be in this sentence. More formally:

$$s(x) = \sum_{j=1}^{\min(n,J)} q^{(j)}$$

11. `worst-token`: The quality of the token that is most likely to be mislabeled determines the sentence's overall quality score, $s(x) = \min\{q_1, q_2, \ldots, q_n\}$. When the token quality estimates $q_j$ are noisy (obtained from limited data), sentence scores that focus only on the smaller token scores corresponding to likely mislabeled tokens can be more effective than those that depend on all tokens' scores

## 4 Results

For evaluation, we apply each sentence scoring method to the given class labels in the CoNLL-2003 named entity recognition dataset (Tjong Kim Sang & De Meulder, 2003). We restrict our attention to the test set, for which all ground truth label errors were identified by Wang et al. (2019) (see Appendix B). We consider two different models to produce per-token predicted probabilities: `bert` (Devlin et al., 2019) and `xlm` (Conneau et al., 2020), and we consider a second variant of the dataset (`unmerged`) with more classes based on additional consideration of `B-` and `I-` entity-prefixes. See Appendix A for all details.

Recall the sentence scores $s(x)$ are used to prioritize which sentences most likely contain label errors. Thus we consider evaluation metrics from information retrieval, which depend on the ranking of sentences induced by $s(x)$ rather than the magnitude of its values. Sentences that contain any mislabeled token are considered true positives when we compute metrics like: **AUROC** (and **AUPRC**) for area under the receiver operating characteristic (and precision-recall) curve. Our third metric, **Lift @ #Errors**, measures how many times more prevalent labels errors are within the top-$T$ scoring sentences vs. all sentences. Here $T$ is the number of true positives ($T = 184$ for `bert` and `xlm`, $T = 186$ for `bert-unmerged`). The Lift metric favors high-precision scores, while AUROC and AUPRC consider both precision and recall, favoring scores capable of detecting a meaningful fraction of all true positives.

Table 1: AUPRC achieved by different sentence scoring (SS) and token scoring (TS) methods described in Section 3. The token score field is left empty for sentence scoring methods that do not rely on token scores.

| SS | TS | bert | xlm | bert-unmerged |
|---|---|---|---|---|
| 1 | | 0.3422 | 0.3412 | 0.3190 |
| 2 | | 0.3087 | 0.3186 | 0.3291 |
| 3 | sc | 0.3740 | 0.3697 | 0.3768 |
| | nm | 0.3702 | 0.3603 | 0.3740 |
| | cwe | 0.3597 | 0.3597 | 0.3609 |
| 4 | sc | 0.3804 | 0.3759 | 0.3901 |
| | nm | 0.3744 | 0.3662 | 0.3822 |
| | cwe | 0.3695 | 0.3602 | 0.3607 |
| 5 | | 0.3131 | 0.3159 | 0.2996 |
| 6 | sc | 0.3022 | 0.3349 | 0.2574 |
| | nm | 0.3066 | 0.3143 | 0.2648 |
| | cwe | 0.2767 | 0.3495 | 0.2572 |
| 7 | sc | 0.3423 | 0.3321 | 0.3229 |
| | nm | 0.3368 | 0.3126 | 0.3221 |
| | cwe | 0.3191 | 0.3380 | 0.3023 |
| 8 | sc | 0.3794 | 0.3559 | 0.3726 |
| | nm | 0.3807 | 0.3533 | 0.3823 |
| | cwe | 0.3519 | 0.3783 | 0.3359 |
| 9 | sc | 0.3383 | 0.3485 | 0.3532 |
| | nm | 0.3776 | 0.3227 | 0.3513 |
| | cwe | 0.3191 | 0.3541 | 0.2980 |
| 10 | sc | 0.3927 | 0.3628 | 0.3614 |
| | nm | 0.3850 | 0.3342 | 0.3603 |
| | cwe | 0.3335 | 0.3620 | 0.3114 |
| 11 | sc | **0.4357** | **0.4021** | **0.4236** |
| | nm | 0.4243 | 0.3963 | 0.3933 |
| | cwe | 0.3215 | 0.3815 | 0.2974 |

Table 1 reports the LED performance of each method and additional evaluation results are in Appendix D. Our results show that `worst-token` (using the `sc` token-score) generally achieves the best LED performance across the three experiments. To most usefully rank sentences for identifying label errors, one should thus account for classifier confidence but not be directly influenced by all tokens' estimated quality (which may be noisy).

## 4.1 Impact of LED on model performance

Finally, we consider the impact that detecting label errors can have on models trained for entity recognition. Here we apply the Confident Learning workflow (Northcutt et al., 2021b), in which we first train a entity recognition model and use its predictions to detect potential label errors, then manually review the estimated errors, and finally train another copy of this same model using the cleaned dataset. Always using the same Bert-NER transformer network (Kanakarajan, 2019), we train three different versions of this model: model A on the original dataset, model B on a version of the dataset where 20% of the sentences were randomly chosen and their labels manually corrected, and model C on a different version of the dataset where 20% of the sentences with the lowest label quality score according to `worst-token` were chosen and their labels manually corrected. In this case, correcting the labels of 20% of the sentences represents a feasible amount of manual labor. To correct the label of a selected sentence, we simply replace its given label in the original dataset by the label provided in CoNLL++. Our results showed that models A, B, C achieved F1-scores (on the same test dataset) of $94.7 \pm 0.39$, $94.98 \pm 0.34$, $95.45 \pm 0.55$, respectively. A t-test demonstrates that model C significantly outperformed model A (p-value = 0.002) and model B (p-value=0.023), while model

B showed a non-significant improvement over model A (p-value=0.062). These findings demonstrate that `worst-token` enables training entity recognition models with significantly better performance compared to those trained on the original dataset or a cleaned version without careful selection of the examples to be corrected. The significant model improvement that results from correcting the labels of particularly-chosen examples emphasizes the importance of targeted label error detection.

# 5    Discussion

This paper studies various ways to estimate the overall labeling quality of sentences in token classification datasets, such that sentences can be prioritized for efficiently quality assurance by data annotation teams. We evaluate 11 methods to score label quality in an entity recognition dataset for which ground truth labels are known and the original labels contain natural errors. A particularly simple `worst-token` approach performs best in this benchmark, and using this method to clean the labels of our training dataset, we are able to significantly improve the accuracy of an entity recognition model without changing any of the modeling code. Our proposed technique is straightforward and can be applied with any token classification model, in order to improve the dataset and then immediately train a better version this same model.

**Limitations of our Work.**   Due to the paucity of token classification datasets in which every label error has been characterized, our evaluations are based on variations of the CoNLL-2003 dataset. While some literature catalogs token-level label errors for other datasets, our manual review uncovered many issues in the raw results. Given that our LED methods are intended to improve real-world datasets, we also want to determine which method is best based on naturally-occurring label errors in real data, rather than evaluating with synthetically introduced errors. The overall effectiveness of our proposed methodology depends on the quality of the model whose predictions are utilized. Worse models make it harder to accurately detect label errors, but imperfect models are key to our scientific study aiming to understand which LED method is the best (with an almost perfect model, almost any reasonable LED method would work fine). As more accurate token classification models are invented, our general `worst-token` method will still remain applicable and its LED abilities will improve alongside the modeling advances (unlike model-specific LED methods).

**Potential Risks.**   The proposed LED methodology makes it easier to discover errors in any token classification dataset, which could potentially be used to maliciously criticize a specific dataset. The proposed algorithmic approach is also unlikely to catch every single error in a dataset and will inevitably output false positives. Thus it is important to review the inferred errors rather than blindly trusting that every flagged sentence actually contains an error. One should also take care not to introduce bias into the dataset when deciding how to handle the label errors.

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

# Appendix

## A  Experiment Details

**Dataset.**   The CoNLL-2003 dataset contains 4 types of named entities: `PER` for persons, `ORG` for organizations, `LOC` for locations, `MISC` for miscellaneous other entities, with `O` being reserved as a label for other types of words that are not named entities. The dataset is in IOB2 format (Ratnaparkhi, 1998), such that all named entities possess an extra `B-` or `I-` prefix, which indicates whether this token is the Beginning of an entity or an Intermediate part of one. We consider this set of 9 classes in `bert` and `xlm`, in which differently prefixed entities are treated as different classes. In the alternative `merged` setting, we consider the `merged` case in where the prefixes are disregarded. In either setting, a sentence is considered "mislabeled" if at least one label of the word-level token differs from original dataset.

**Models.**   For producing predicted probabilities to input into our label quality scoring methods, all models were only trained on the training set of CoNLL-2003, and we use them to produce held-out predictions for the test set. No model has access to the test set examples during training, nor any ground-truth label errors at any point in our evaluation.

We consider three different settings in our experiments, described below in the corresponding order: `bert-unmerged`, `bert`, `xlm`. The first model we consider is a pre-trained `bert-base-NER` Transformer network, which has been found to be an effective token classifier for CoNLL-2003 including the `B-` and `I-` entity-prefixes as additional classes (for a total of 9 classes) (David S.). We conduct two additional experiments to verify whether the label error detection results with this model remain consistent in other settings. First, we omit the `B-` and `I-` prefixes such that we only focus on more severe error types, such as `LOC` vs. `ORG` rather than `B-LOC` vs. `I-LOC` label substitutions (which may be of less interest). With this reduced set of entities, we still use the same Bert network, which now predicts amongst a fewer set of 5 classes for each token. Finally, to examine how our methods work when applied with different models, we also obtain a different set of model-predicted probabilities using another pre-trained XLM network: `xlm-roberta-large-finetuned-conll03-english` (The team at Hugging Face). Given that the model outputs predictions in the IOB format (Ramshaw & Marcus, 1999), we again consider the reduced set of 5 classes.

**Data Processing.**   Before applying our label error detection methods, we first preprocess the original CoNLL-2003 dataset. Sentences less than or equal to 1 character, and sentences containing `"#"` are excluded. The latter because `"#"` is a special character reserved by many token classification models to represent subword tokens. We construct each sentence by joining the tokens separated by space, and perform some minor cleanup to ensure that the sentence follows writing conventions (such as no space before comma, and no space after open parenthesis). We convert all-caps tokens into lowercase except for the first character (i.e. `JAPAN -> Japan`), because token classification models tend to partition all-caps tokens into multiple subword-level tokens. This can sometimes result in some undesirable behaviors, such as converting `USA` to `Usa`, or `NBA` to `Nba`. We believe that the benefits outweigh the costs for CoNLL-2003, due to the prevalence of article headlines in which most, if not all, tokens in the sentence are all-caps.

Next, we use a pre-trained token classification model to obtain the model-predicted probabilities. Modern Transformer models first tokenize the sentence into multiple subword-level tokens, possibly different from the given tokens individually labeled in the original dataset. These "subword-level" tokens are typically smaller units than word-level tokens, and the trained model outputs a probability distribution over possible classes for each such token.

Here we reduce these probabilities from subword-level to word-level tokens, so that we can evaluate the given labels as outlined in the main text. Consider the sentence:

```
Minnesota Timberwolves (MIN)
```

where the given tokens from original dataset are `["Minnesota", "Timberwolves", "(", "MIN", ")"]`, and the model tokenizes the sentence into subword-level tokens: `["Minnesota", "Timber", "wolves", "(MIN)"]`[2]. Let $\mathbf{p}^{(i)} = [p_1^{(i)}, p_2^{(i)}, \ldots, p_K^{(i)}]$ denote the model-predicted probabilities of the $i$th subword-level token, where $K$ is the number of possible classes. To obtain the predicted probability distribution for `"Timberwolves"` (used to evaluate its given label in the original dataset), we take the average of the (model-estimated) probabilities for `"Timber"` $\mathbf{p}^{(2)}$, and `"wolves"` $\mathbf{p}^{(3)}$. We assign the probability for `"(MIN)"` $\mathbf{p}^{(4)}$ directly to `"("`, `"MIN"` and `")"`. We used similar strategies for other model-tokenization-induced discrepancies in the data.

We also experimented with alternative probability-pooling methods for adjacent subword-level tokens, such as: a weighted average (with weights proportional to the number of characters in each subword-level token, such that longer strings receive a higher weight), or considering only the predicted probabilities for first subword-level token. Both of these variants produced little differences in their LED performance from the average-pooling technique we employ for the results shown in this paper.

**Hyperparameter Settings.** 5 of the sentence scoring methods considered in this paper contain a hyperparameter (but not our proposed `worst-token` method). To ensure these other methods are not disadvantaged vs. `worst-token`, we tried many different values of their hyperparameter and report results for the best hyperparameter value. See Appendix C for descriptions of some additional methods not introduced in the main text. The values considered for each method are as follows, with the selected hyperparameter in **bold**.

- `product`: $10^{-1}, 10^{-1.5}, 10^{-2}, 10^{-2.5}, \mathbf{10^{-3}}$

- `worst-token-min-alt`: $0.01, 0.02, 0.03, 0.04, 0.05, 0.06, 0.07, 0.08, 0.09, \mathbf{0.1}$

- `worst-token-softmin`: $10^{-1}, 10^{-1.1}, 10^{-1.2}, 10^{-1.3}, 10^{-1.4}, \mathbf{10^{-1.5}}$

- `expected-bad`: $\mathbf{2}, 3, 4$

- `expected-alt`: $\mathbf{2}, 3, 4$

Note that `expected-bad` and `expected-alt` with a hyperparameter value of 1 are identical to `worst-token`, so this value is excluded from our hyperparameter sweep.

**Compute Details.** Our experiments relied on already trained models, in order to demonstrate that our proposed error-detection methodology works with arbitrary models. These experiments can be run in under an hour on a 20 CPU Linux virtual machine and do not require a GPU (which would however be required to train the models as well).

**Artifacts.** Our work uses these artifacts:

- CoNLL-2003 shared task: collected by Tjong Kim Sang & De Meulder (2003) and consists of Reuters news stories between August 1996 and August 1997. It is intended for natural language processing research (Apache 2.0 License). The dataset does not include information that uniquely identifies individual people nor does it contain offensive language. The dataset includes 20744 sentences, which are split in train/validation/test parts (14041/3250/3453, respectively). A small number of sentences are excluded from our experiment, as discussed in Appendix A.

- CoNLL++: collected by Wang et al. (2019), in which 5 human experts were hired to manually correct mislabels in the original CoNLL-2003 corpus. It is intended for natural language processing research (Apache 2.0 License).

---

[2]Different models may result in different tokenization.

- `bert-base-NER`: trained by David S. on the CoNLL-2003 dataset, this model is intended for named entity recognition applications (MIT License).

- `xlm-roberta-large-finetuned- conll03-english`: trained by Conneau et al. (2020), this is a large multi-lingual language model trained on 2.5TB of filtered CommonCrawl data, and subsequently fine-tuned on the CoNLL-2003 dataset (Apache 2.0 License).

# B  Ground Truth Label Errors in CoNLL (via CoNLL++)

Wang et al. (2019) manually corrected the entire test set of CoNLL-2003, discovering 186 sentences (5.38% of the original data) that contain at least one token label error. Through extensive manual inspection, we comprehensively verified that basically all CoNLL-2003 label errors (in test set) have been fixed by Wang et al. (2019), and their proposed corrections are reliable (we did not find false positives or negatives). Wang et al. (2019) named their corrected version `CoNLL++`, which we consider as a source of ground truth to validate candidate label errors for this study.

Reiss et al. (2020) also proposed a corrected set of labels for CoNLL-2003, identified via semi-supervised algorithms and limited manual label verification. However, close examination of their proposed corrections reveals that many of them are fundamentally incorrect and many real CoNLL label errors (found by Wang et al. (2019)) were not properly identified/fixed by Reiss et al. (2020). For example, consider the first sentence from the test set:

Soccer - Japan get lucky win, China in surprise defeat.

In the original CoNLL-2003 dataset, `Japan` is labeled `LOC`, and `China` is labeled `PER` (this happens to be a label error, it should be `LOC` instead). Here, we omit the `B-` and `I-` prefix. In the dataset corrected by Reiss et al. (2020), `Japan` and `China` are labeled `ORG`, both of which are incorrect. In addition, consider another sentence from the test set:

...is not for a stronger dollar either," said Sumitomo's Note.

In the dataset corrected by Reiss et al. (2020), `"Note"` is labeled as `PER`, while the correct label should be `O`. The token is also labeled incorrectly in the original dataset, but is corrected by Wang et al. (2019). Overall comparing against the high-quality (comprehensively manually verified) label corrections in `CONLL++`, we found the (mostly algorithmically) CoNLL-corrected dataset from Reiss et al. (2020) contains many corrections which are not actually valid (we estimate around 8% of their proposed corrections are wrong). Hence, we opted not to base any evaluations on this dataset from Reiss et al. (2020). This additionally highlights the difficulty of algorithmically correcting an entire dataset's labels, which is why we focus on label error detection in this work, developing methods that enable human reviewers to quickly find and fix the label errors.

In Table 2, we present the estimated label noise matrix between the original CoNLL-2003 dataset vs. CoNLL++ on the token-level. The $i$th row and $j$th column of the table below represents the percentage of tokens that are labeled $i$ in CoNLL++, and mislabeled as $j$ in the original CoNLL-2003 dataset.

# C  Variants of our `worst-token` method

For completeness, we also study some minor variants of our proposed methodology. Using the notation in Section 3, we consider the following alternative methods to produce a sentence score $s(x)$ for sentence $x$:

- `worst-token-min-alt`: We add quality-score penalty $d$ for tokens flagged as likely label errors by Confident Learning (Northcutt et al., 2021b), and then consider the worst token based on the penalized quality scores. More formally:

$$s(x) = \min_i(q_i + d \cdot b_i)$$

Table 2: Estimated label noise matrix between CoNLL_2003 vs. CoNLL++ on the token-level.

|  | O | B-MISC | I-MISC | B-PER | I-PER | B-ORG | I-ORG | B-LOC | I-LOC |
|---|---|---|---|---|---|---|---|---|---|
| O | - | 0.01% | 0.02% | 0.01% |  | 0.01% |  |  |  |
| B-MISC | 5.39% | - |  |  |  | 0.14% |  | 2.49% |  |
| I-MISC | 18.90% | 1.57% | - |  |  |  |  |  |  |
| B-PER | 0.49% | 0.12% |  | - |  | 0.06% |  | 0.19% |  |
| I-PER | 0.17% |  |  | 0.43% | - |  |  |  | 0.09% |
| B-ORG | 0.82% | 1.17% |  | 0.18% |  | - |  | 1.87% |  |
| I-ORG | 3.18% |  | 0.68% |  | 0.34% | 0.34% | - | 0.23% | 0.80% |
| B-LOC | 0.91% | 0.49% |  | 0.18% |  | 0.43% | 0.06% | - |  |
| I-LOC | 2.70% |  | 0.77% |  |  |  | 0.39% |  | - |

Recall $i$ ranges over the tokens in sentence $x$. We experimented with different values of constant hyperparameter $d$. When $d \geqslant 1$, sentences that contain at least one token flagged as a likely label error by Confident Learning are completely separated from the remaining sentences. In other words, if a sentence does not include any flagged tokens, it is guaranteed to rank below all of the sentences with at least one flagged token.

- `worst-token-softmin`: Instead of considering only the worst token's quality score, we softly consider all other token's quality-score (to a lesser degree) as well in the overall sentence score. More formally:

$$s(x) = \langle \mathbf{q}, \mathrm{softmax}_t(\mathbf{1} - \mathbf{q}) \rangle$$

Here $\langle \cdot, \cdot \rangle$ denotes the inner dot product. The temperature of the softmax $t$ is a constant hyperparameter we tried different values for. Settings of $t$ closer to 0 make this approach converge to our `worst-token` method, whereas this approach converges to the `average-quality` method with large values of $t$.

We evaluate the Lift, AUROC and AUPRC of our proposed method and its variants for comparison. The setup is the same as before, where we report results from the variants with the best hyperparameter settings we could find (to ensure they are not unfairly disadvantaged against `worst-token`). These settings were $d = 0.1$ for `worst-token-min-alt` and $t = 10^{-1.5}$ for `worst-token-softmin`. The best performing scoring method is highlighted in **bold**.

Table 3: Lift @ #Errors for our proposed method `worst-token` and its variants.

| Sentence score | Token Score | bert | xlm | bert-unmerged |
|---|---|---|---|---|
| `worst-token` | sc | 9.02 | 8.71 | 8.83 |
|  | nm | 9.02 | 8.71 | 8.73 |
|  | cwe | 7.40 | 7.90 | 6.35 |
| `worst-token-min-alt` | sc | 9.02 | 8.71 | 8.63 |
|  | nm | 9.02 | 8.71 | 8.73 |
|  | cwe | 7.90 | 7.70 | 7.83 |
| `worst-token-softmin` | sc | 8.92 | **8.82** | 8.53 |
|  | nm | **9.83** | 8.51 | **9.12** |
|  | cwe | 7.50 | 8.00 | 6.35 |

The `worst-token-softmin` variant provides a "second chance" for other mislabeled tokens' (presumably) low quality scores (e.g. the second lowest) to be considered in the overall sentence quality score. For example, consider two sentences with token quality scores $[0.01, 0.99]$ and $[0.011, 0.02]$, respectively. `worst-token` will assign a lower score to the first sentence, but if the first token turns out not to be a label error, the sentence will become a false positive. On the other hand, `worst-token-softmin` also considers the second lowest token quality score, and assigns a lower score for the second sentence. In this case, it may be more likely for at

Table 4: AUPRC for our proposed method `worst-token` and its variants.

| Sentence score | Token Score | bert | xlm | bert-unmerged |
|---|---|---|---|---|
| `worst-token` | sc | 0.4357 | 0.4021 | **0.4236** |
| | nm | 0.4243 | 0.3963 | 0.3933 |
| | cwe | 0.3215 | 0.3815 | 0.2974 |
| `worst-token-min-alt` | sc | **0.4408** | 0.4019 | 0.4181 |
| | nm | 0.4238 | 0.3965 | 0.3945 |
| | cwe | 0.3643 | 0.3819 | 0.3081 |
| `worst-token-softmin` | sc | 0.4402 | **0.4063** | 0.4171 |
| | nm | 0.4388 | 0.3980 | 0.4065 |
| | cwe | 0.3185 | 0.3799 | 0.2937 |

Table 5: AUROC for our proposed method `worst-token` and its variants.

| Sentence score | Token Score | bert | xlm | bert-unmerged |
|---|---|---|---|---|
| `worst-token` | sc | 0.9058 | **0.9141** | **0.8905** |
| | nm | 0.9059 | 0.9134 | 0.8852 |
| | cwe | 0.8996 | 0.9121 | 0.8834 |
| `worst-token-min-alt` | sc | **0.9066** | 0.9140 | 0.8897 |
| | nm | 0.9058 | 0.9135 | 0.8861 |
| | cwe | 0.9027 | 0.9119 | 0.8872 |
| `worst-token-softmin` | sc | 0.9026 | 0.9074 | 0.8878 |
| | nm | 0.9040 | 0.9064 | 0.8830 |
| | cwe | 0.8962 | 0.9046 | 0.8808 |

least one of the tokens in the second sentence to be mislabeled vs. only the first token in the former sentence. Hence `worst-token-softmin` results in attains better Lift @ #Errors than `worst-token`. However being more dependent on all tokens' label quality scores makes `worst-token-softmin` more sensitive to estimation error than `worst-token`. Thus `worst-token-softmin` does not necessarily produce a better overall ranking of the sentences in terms of AUPRC/AUROC.

`worst-token-min-alt` utilizes additional information beyond `worst-token`, which is estimated via Confident Learning for flagging likely label errors. Tables 4 and 5 show that the additions in `worst-token-min-alt` and `worst-token-softmin` can sometimes provide slight benefits to `worst-token`, but we do not find the performance gains to be significant enough to warrant the additional complexity (and hyperparameters) of these variants.

# D   Additional Results

Tables 6 and 7 provide additional sentence scoring evaluations under the Lift and AUROC metrics. Note the token score field is left empty for sentence scoring methods that do not rely on token scores. Figure 1 shows the precision for label error detection achieved by a subset of our sentence scoring methods.

Table 6: Lift @ #Errors for varying sentence and token scoring methods.

| Sentence score | Token Score | bert | xlm | bert-unmerged |
|---|---|---|---|---|
| predicted-difference | | 7.19 | 6.59 | 7.14 |
| bad-token-counts | | 7.30 | 6.89 | 8.03 |
| bad-token-counts-avg | sc | 7.80 | 7.09 | 8.13 |
| | nm | 7.60 | 7.40 | 8.13 |
| | cwe | 7.30 | 6.89 | 7.24 |
| bad-token-counts-min | sc | 7.80 | 7.09 | 8.23 |
| | nm | 7.60 | 7.40 | 8.13 |
| | cwe | 7.30 | 6.89 | 7.24 |
| good-fraction | | 5.88 | 6.18 | 5.45 |
| penalize-bad-tokens | sc | 5.98 | 6.18 | 5.55 |
| | nm | 5.98 | 6.18 | 5.65 |
| | cwe | 6.38 | 6.08 | 5.75 |
| average-quality | sc | 5.07 | 5.88 | 4.76 |
| | nm | 5.07 | 5.98 | 4.66 |
| | cwe | 5.37 | 6.18 | 5.26 |
| product | sc | 7.30 | 6.99 | 7.24 |
| | nm | 7.60 | 7.40 | 7.64 |
| | cwe | 6.79 | 7.09 | 7.24 |
| expected-bad | sc | 6.99 | 6.48 | 6.84 |
| | nm | 6.69 | 6.48 | 7.24 |
| | cwe | 6.59 | 6.08 | 7.34 |
| expected-alt | sc | 6.99 | 6.48 | 6.94 |
| | nm | 6.89 | 6.59 | 7.24 |
| | cwe | 7.09 | 6.89 | 7.34 |
| worst-token | sc | **9.02** | **8.71** | **8.83** |
| | nm | 9.02 | 8.71 | 8.73 |
| | cwe | 7.40 | 7.90 | 6.35 |

Table 7: AUROC for varying sentence and token scoring methods.

| Sentence score | Token Score | bert | xlm | bert-unmerged |
|---|---|---|---|---|
| predicted-difference | | 0.8639 | 0.8633 | 0.8559 |
| bad-token-counts | | 0.8188 | 0.8421 | 0.8114 |
| bad-token-counts-avg | sc | 0.8934 | 0.9033 | 0.8718 |
| | nm | 0.8932 | 0.9030 | 0.8702 |
| | cwe | 0.8902 | 0.9012 | 0.8725 |
| bad-token-counts-min | sc | 0.9026 | 0.9106 | 0.8885 |
| | nm | 0.9025 | 0.9102 | 0.8854 |
| | cwe | 0.9023 | 0.9091 | 0.8892 |
| good-fraction | | 0.8147 | 0.8393 | 0.8049 |
| penalize-bad-tokens | sc | 0.8151 | 0.8396 | 0.8053 |
| | nm | 0.8151 | 0.8396 | 0.8058 |
| | cwe | 0.8162 | 0.8401 | 0.8064 |
| average-quality | sc | 0.8553 | 0.8895 | 0.8079 |
| | nm | 0.8560 | 0.8894 | 0.8068 |
| | cwe | 0.8578 | 0.8913 | 0.8305 |
| product | sc | 0.8900 | 0.8647 | 0.8811 |
| | nm | 0.8905 | 0.8646 | 0.8784 |
| | cwe | 0.8870 | 0.8683 | 0.8783 |
| expected-bad | sc | 0.8946 | 0.9026 | 0.8724 |
| | nm | 0.8948 | 0.9017 | 0.8705 |
| | cwe | 0.8922 | 0.9033 | 0.8778 |
| expected-alt | sc | 0.8963 | 0.9060 | 0.8776 |
| | nm | 0.8972 | 0.9044 | 0.8759 |
| | cwe | 0.8950 | 0.9048 | 0.8825 |
| worst-token | sc | 0.9058 | **0.9141** | **0.8905** |
| | nm | **0.9059** | 0.9134 | 0.8852 |
| | cwe | 0.8996 | 0.9121 | 0.8834 |

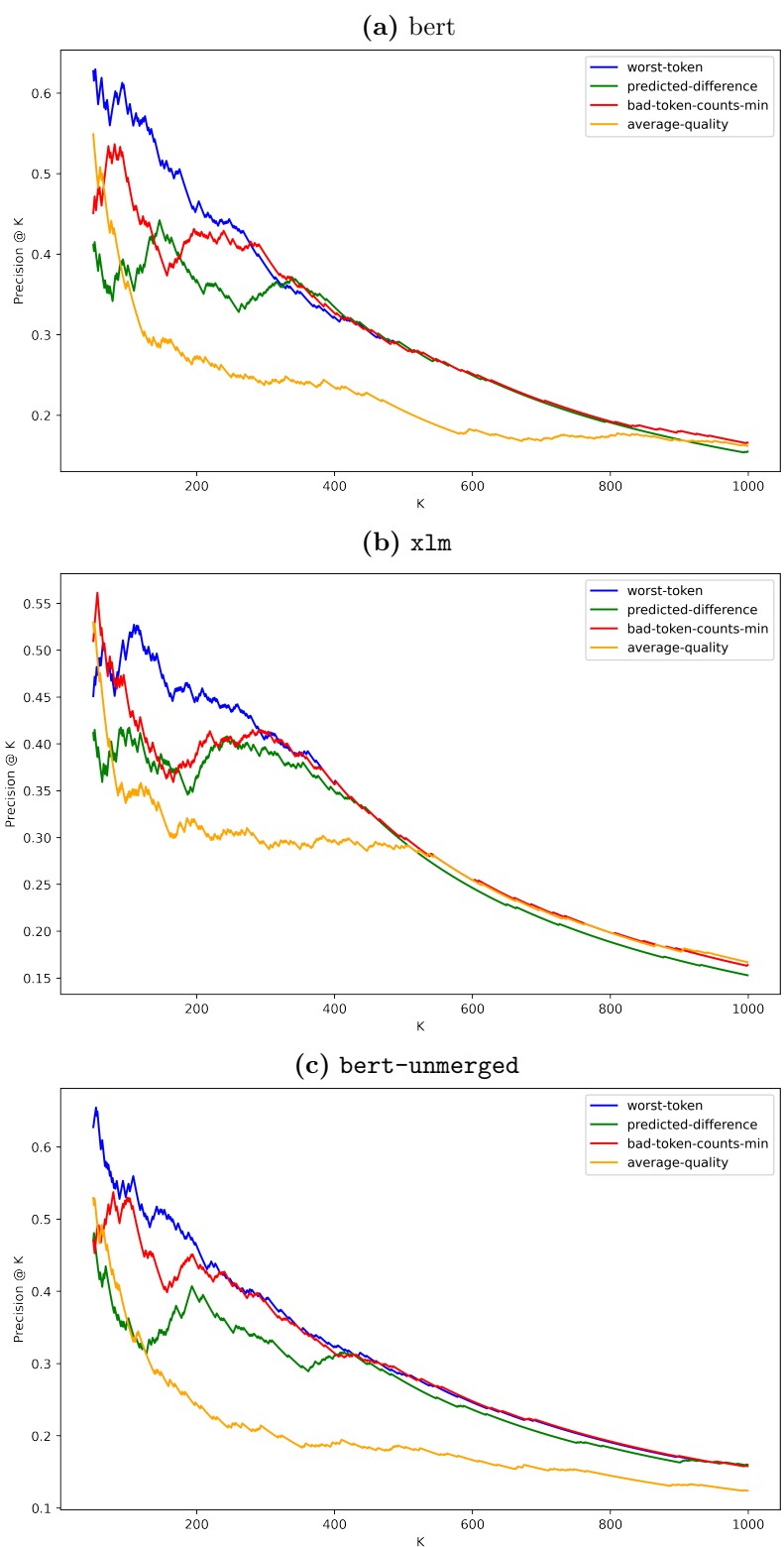

Figure 1: Precision @ K for detecting sentences that contain a label error via our `worst-token` sentence scoring method and three others (over different values of $K$). Across the 3 experiments, `worst-token` consistently detects sentences containing label errors with higher overall precision.

# E   Top Label Issues using `worst-token` method

Based on the given labels and out-of-sample predicted probabilities, `worst-token` can quickly help us identify label issues in the dataset. Here we list the top 10 label issues, ie. sentences with the lowest quality score, and display their given and predicted labels. Given that `O` and `MISC` can sometimes be ambiguous, they are excluded in the following examples. Similarly, we omit the `B-` and `I-` tags (see Appendix A).

1. Little change from today's weather expected.
   Given label: `PER`, Predicted label: `O`

2. Let's march together...
   Given label: `LOC`, Predicted label: `O`

3. Nastja Rysich (Germany) 3.75
   Given label: `LOC`, Predicted label: `O`

4. ...from the Moslem, Arabised north.
   Given label: `LOC`, Predicted label: `O`

5. Mayor Antonio Gonzalez Garcia...
   Given label: `PER`, Predicted label: `O`

6. Spring Chg Hrw 12pct Chg White Chg
   Given label: `LOC`, Predicted label: `O`

7. ...Prince Rainier told Reuters
   Given label: `PER`, Predicted label: `O`

8. Danila 28.5 16/12 Caribs/ up W224 Mobil.
   Given label: `O`, Predicted label: `LOC`

9. ...next Wednesday.
   Given label: `ORG`, Predicted label: `O`

10. ...Sale Limits US / UK / JP / FR
    Given label: `LOC`, Predicted label: `O`

More than half of the potential label issues correspond to tokens that are incorrectly labeled. As shown above, some examples are ambiguous and may require more thoughtful handling, and some edge cases are simply punctuations such as "/".

# F   Label Error Detection on the Token-level

While our main focus was identifying sentences likely to contain a mislabeled token (given that reviewing an individual token's label anyway requires reading the broader context contained in the sentence), here we examine how accurately different estimates identify the erroneous tokens themselves. Recall from Section 3, we can use predicted class probabilities for each token from our trained model to compute a label quality score for each token $q_i$. We study three different label quality-scores (`sc`, `nm`, `cwe`) considered by Kuan & Mueller (2022) to identify label errors in multi-class classification tasks.

After computing these scores for every token in the CoNLL-2003 test set (unmerged, ie. keeping the `B-` and `I-` tags), we evaluate token-level label error detection by sorting the tokens according to these scores (regardless

of which sentence the tokens belonged to) and report the precision/recall for detecting which of these tokens were actually mislabeled or not (utilizing CoNLL++). Table 8 and Figure 2 contain the same precision/recall metrics described in Section 4, now computed at the token-level rather than sentence-level. `sc` and `nm` give the best performance (according to AUPRC or Lift metrics) for detecting which tokens are mislabeled. Unlike standard multi-class classification datasets where Kuan & Mueller (2022) found `cwe` to be an effective label quality score to flag out-of-distribution examples for which no class label is appropriate, CoNLL-2003 and other entity recognition datasets contain an "other" class that explicitly accounts for tokens which do not belong to one of the classes of interest.

Table 8: Evaluating token-level label quality scores $q_i$ on the CoNLL-2003 test set.

|  | AUPRC | AUROC | Lift @ # errors |
|---|---|---|---|
| `sc` | 0.3483 | 0.9545 | 62.487 |
| `nm` | 0.3296 | 0.9552 | 66.362 |
| `cwe` | 0.2293 | 0.9355 | 46.986 |

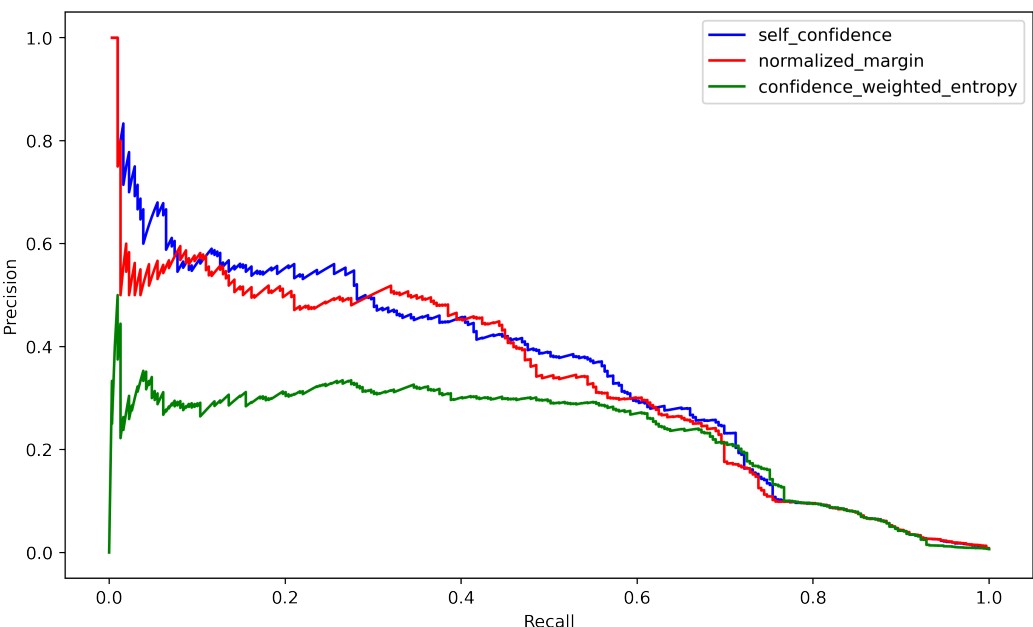

Figure 2: Precision-Recall curve for three different label quality scoring methods on the token level.

