# OpenReview forum: "Detecting Label Errors in Token Classification Data "
_TMLR — Rejected by TMLR_

### Review · Reviewer_gjWf · 2023-07-10

**Summary Of Contributions:**

The paper proposes a heuristic to detect erroneous labels in NER (and perhaps other sequence tagging).
It suggests and tests many heuristics in order to choose the one that works best.
The work is very empirical in nature and evaluates 11 formulas over model scores (and 3 normalization methods).

**Audience:**

Yes

**Claims And Evidence:**

Yes

**Requested Changes:**

The introduction does not give a good approximation of what is going to be discussed. The paper tests many options and variations but in the introduction it seems like it only tries one heuristic and shows it works.

Not critical:
is token classification a common term? I find sequence tagging to be the one I am familiar with.

Entity-disjoint is left for the reader to understand, despite being used several times. The idea can be explained in another sentence or so.
Similarly, the mislabeled tokens (b) algorithm, could use another sentence explaining how it chooses the adaptive thresholds.

**Strengths And Weaknesses:**

Strength

The paper is clear and easy to follow.

The paper tests various options for identifying mislabelings.

Weaknesses

The scope of the contribution is quite narrow.

As a paper that proposes many heuristics one would expect the method to be tested in many settings. Otherwise, the heuristic may not be a general finding but just one that fits the current scenario.

I wonder why larger models weren't used, and why not deberta rather than bert and roberta.

Moreover, the language the paper works on is never mentioned, as far as I could see, but the authors use XLM, a multilingual model and bert a monolingual model, this a very surprising choice to me.

Some understanding of what makes one heuristic better than another, manual analysis etc. might have given more room for future improvements and more to the feeling that the leading method indeed provides meaningful improvements.

---

### Review · Reviewer_uivY · 2023-08-22

**Summary Of Contributions:**

This paper compares different features (see section 3) for cleaning token-level label error and found out that a statistics called "worst token" is better than other proposed statistics. The proposed statistics aggregated several subcomponents that are built around the model predicted probabilities (e.g. margin between the most confident class and the rest). In constructing the statistics, the paper also introduces some additional hyperparameters in combining them. The authors evaluate the proposed method on the named entity recognition task and on the CoNLL 2003 dataset.

**Audience:**

Yes

**Broader Impact Concerns:**

I do not see ethical concerns with this work.

**Claims And Evidence:**

Yes

**Requested Changes:**

I think the manuscript would be benefitted from adding additional experiments on other datasets and token-level classification tasks beyond named entity recognition.

**Strengths And Weaknesses:**

Strengths
- This paper is easy to understand and attack a well-defined problem

Weakness
- the experiments are only done on the CoNLL-2003 dataset and I am not convinced that the method generalizes to a broader spectrum of problems.

---

### Review · Reviewer_ev3v · 2023-09-02

**Summary Of Contributions:**

The paper considers the problem of measuring the quality of labels in token-level classification datasets. Token-level
classification is defined in this paper as the task of classifying individual tokens in natural language sentences, like in Named
Entity Recognision (NER), for example. The authors argue that this is an important problem in modern machine learning
datasets, significantly affecting both the performance of models trained on such noisy data, and the evaluation results
computed over such noisy data.

The paper considers 11 different methods that have been previously published to evaluate the label quality in existing
data, and evaluates their performance on the CoNLL-2003 dataset. Specifically, the authors study 3 methods for
evaluating the per-token labels, individually and independently, and 8 methods for combining the token-level qualities
to derive sentence-level qualities. The motivation for the latter is that the selected "bad" examples will be sent to
humans for annotation and in most cases human annotators would have to look at the full sentence to determine the
correct per-token labels. Thus, the end result of this process would ideally be full sentences to be relabeled.

They show that one of the studies methods significantly outperforms the rest and results in practical gains when used
to relabel "bad" examples and retrain a classifier on them.

**Audience:**

Yes

**Claims And Evidence:**

No

**Requested Changes:**

I would strongly recommend that you consider addressing weaknesses 2 and 3 from the list I provided above. It would certainly significantly improve your paper and its reception by the community. It would also be helpful in getting me closer to recommending for acceptance. Though, I do not feel like I can recommend yet because the presented work feels more like a re-implementation of ideas that have been published elsewhere without providing any additional insights into them.

**Strengths And Weaknesses:**

### Strengths

1. Extensive list of studied methods.
2. The paper is well-written and easy to follow.
3. Even though the problem as stated considers token-level human-provided noisy labels, it is important and pertinent
   to current trends in the machine learning community, where researchers are attempting to use Large Language Models (LLMs)
   to obtain noisy labels over unlabeled data and train on them. Solutions found for the problem presented in the paper
   would apply in this setting as they could be used to detect label quality issues in such auto-generated data and
   potentially prompt relabeling (even using the same underlying LLM with additional information provided in the prompt).

### Weaknesses

1. Perhaps the biggest weakness I see is that this feels more like a short paper than a regular journal paper. Specifically,
   it feels like something along the lines of a short EMNLP findings paper would be a more appropriate venue.
2. There are a lot of methods that are studied but unfortunately they are just enumerated and no discussion is provided
   on intuition behind them. For example, what is the purpose of the "max" term in "predicted-difference"? Or what are
   the trade-offs between using "bad-token-counts-avg" vs "bad-token-counts-min"? What should one do if they care more
   about recall than precision? What if some types of tokens are way more important than others and the class distribution
   is very imbalanced? Given that this paper is a study/review of these approaches, it is important to include some
   discussion on the trade-offs and intuitions of the different approaches.
3. While there is an extensive list of methods that are being studied, the experiments section is lacking. Only a single
   small and old dataset is used for experiments and evaluation. It is important to include experiments using more
   and larger datasets. Also, the results of Table 1 are a bit hard to interpret. What do these numbers mean? Are they
   significant? Relatedly, the authors could consider a more modern setup like generating a
   dataset that is automatically labeled using an LLM, applying their approach and then using the LLM to relabel bad
   quality examples, and eventually fine-tuning something like BERT over the original dataset and the relabeled one,
   similar to their current model B vs model C experimental setup.

---

### Decision · Action_Editor_o9c7 · 2023-11-21

**Recommendation:** Reject

**Comment:**

The reviewers recognize the importance of the problem tackled by this paper, but are not convinced by the experiments, specifically the lack of more than one dataset and a limited set of models makes it hard to tell if the method will generalize well. Reviewers also mentioned that the paper does not provide any background or rationale for the heuristics tested (both pre-existing and new ones) and the authors do not perform any analysis on their results. Overall, this makes the paper below par for publication in TMLR in its current state, but I encourage the authors to flesh out their experiments and background sections in order to strengthen the paper.

**Audience:**

This work will likely be of interest to the general machine learning community since data labeling and quality are critical to model training.

**Claims And Evidence:**

This paper proposes techniques to detect mislabeled examples in annotated datasets for token classification. The authors propose simple methods that seem to be competitive with pre-existing approaches on the CoNLL 2003 dataset. Reviewers note however that the experiments are performed with a limited number of models, on just one dataset that is 20 years old, and with no further analysis to inform future work in this space.